# Calculating the Surface Layer Thickness and Surface Energy of Aircraft Materials

Victor M. Yurov [1], Vladimir I. Goncharenko [2], Vladimir S. Oleshko [2] and Anatoly V. Ryapukhin [2,*]

1    Academician Y.A. Buketov Karaganda State University, Universitetskaya Street 28, Karaganda 100028, Kazakhstan
2    Moscow Aviation Institute, Volokolamskoe Highway 4, 125993 Moscow, Russia
*    Correspondence: ryapukhin.a.v@mail.ru

**Abstract:** The surface layer determines the physical properties of aviation materials and, based on these properties, the calculation of surface energy anisotropy can be implemented. Moreover, the value of the surface energy determines the service time and the destruction of aircraft structures surface layer, while the surface layer thickness determines the distance at which this process usually takes place. In this work, a new atomically smooth crystal empirical model is built without considering the surface roughness. This model can be used to theoretically predict the surface energy anisotropy and surface layer thickness of metals and other compounds, in particular the aviation materials. The work shows that the surface layer of an atomically smooth metal, like other compounds, consists of two nanostructured layers: d(I) and d(II). Having sufficient accuracy, the proposed model would allow the prediction of aviation materials performance properties without the need for ultrahigh vacuum or other complicated theoretical methods to analyze the surfaces of nanosystem atomic structures.

**Keywords:** aerospace industry; aviation materials; surface layer thickness; surface energy anisotropy; nanostructure





## 1. Introduction

Up to date modeling structure, structure–property coupling, and qualitative prediction of new physical and physicochemical properties of crystalline and nanostructured substances and materials is still considered one of the most important research directions in the field of new aviation materials creation. Gibbs considered the surface layer as a geometric surface without thickness, while Van der Waals, Guggenheim and Rusanov considered it as a layer of finite thickness [1]. According to modern concepts, a surface layer is imagined as an ultra-thin envelope of unknown size and in thermodynamic equilibrium with a crystalline substrate, where its properties, structure, and composition are different from the bulk layers. In this work, we present an original model that allows theoretically determination of the surface layer thickness of metals, dielectrics, semiconductors, alloys, minerals, etc., as well as aviation materials. This model is based on two fundamental parameters: molar volume and the density of elements. The model can be easily used for computer simulation for any cases that arise in materials science without experimental verification. For example, the model allowed us, for the first time, to determine the thickness of a pure metal's surface layer, which ranges from 2 to 5 nm for a nanostructure and is 135 nm for C96 fullerenes, which obviously exceeds the size of a nanostructure (~100 nm).

In order to clearly understand the proposed model and apply it to aviation and other materials, we will explain the bases upon which this model is built starting from the surface of solid bodies down to the surface layer thickness for aircrafts:

- surface of solid bodies: this surface represents the layer through which the body interacts with the outside world (through its nanostructure to be more precise);

- determination of the solid bodies surface energy: this is the energy characteristic of the surface, nanostructure, and solid body, which determines all the physicochemical processes taking place on it up to its destruction;
- features of the aviation materials' surface layer: they are related to the difference between the solid body ideal surface and its real surface with roughness;
- surface layer thickness of pure metals: this is the thickness for which the layer is considered a nanostructure (3–5 nm);
- surface layer thickness and surface energy of aviation materials: these are aluminum and nickel alloy materials that have a surface layer thickness of approximately 6–9 nm. However, for aircraft where the coating is in the form of metal oxides, the surface layer thickness is about 70–90 nm. In this case, the thickness of the oxide layer of these metallic materials can range from 5 to 90 nm depending on the modes of their processing.

After that, the paper presents studies showing the role of the thickness of the surface layer in the formation of the magnitude of the surface energy, on which many properties of aviation materials depend.

## 2. Materials and Methods

### 2.1. Surface of Solid Bodies

The studies of solid bodies surface began in the 1960s when scientists mastered the art of making ultra-high vacuum chambers in which the pressure is within $3 \times 10^{-11}$–$3 \times 10^{-10}$ torr, while the temperature is up to 10 K. Consequently, it became possible to create atomically smooth and atomically clean crystal surfaces [1]. Modern methods for studying solid bodies surface are divided into the following groups [2,3]: low-energy electron diffraction (LEED), in which only the layers of atoms closest to the surface (~0.5 nm) take part in the formation of LEED pattern; grazing incidence X-ray diffraction (GIXRD), where the angle of incidence is equal to or less than the critical angle of the total internal reflection; Auger electron spectroscopy (AES), where the depth of the analyzed layer varies from ~0.5 nm (at energy of 50 eV) to ~2 nm (at Auger electrons energy of 500 eV); electron energy loss spectroscopy (EELS), where the layer depth is approximately 1 nm; X-ray photoelectron spectroscopy (XPS) [4], where the depth of the analyzed layer is (0.5–2.5) nm for metals and (4–10) nm for organic and some polymeric materials; ion scattering spectroscopy (ISS), which is used to determine the surface composition and structure; secondary ion mass spectrometry (SIMS), which analyzes the sputtered substance where the thickness of the analyzed layer depends on the primary ions energy and can reach ~5 nm; frustrated total internal reflection (FTIR), in which the layer thickness is about ~1 μm for high-molecular solids; transmission electron microscopy (TEM) [5], which cannot exceed 100 nm and usually ranges from 20 nm to 30 nm; scanning electron microscopy (SEM), which is used to study the supramolecular formations morphology in crystalline and amorphous polymers; scanning tunneling microscopy (STM), where the tip is brought to the surface at <1 nm distance and the surface structure picture can be obtained at the atomic level; and atomic force microscopy (AFM), where the surface under study relief is formed either in the constant height or constant force mode and the picture of the surface structure can be also obtained at the atomic level. These methods for studying the surface of solid bodies are laborious, not only because of the required ultrahigh vacuum equipment, but also due to the technology needed to cultivate ultrapure single crystals. Moreover, the creation of ultra-high vacuum is necessary to use the above-listed methods. This fact can, in turn, add complexity to finding ultra-pure metals [6,7] in order to obtain atomically smooth surfaces. On the other hand, the theoretical methods for studying the nanosystem's atomic structure are well-known: the method of classical (empirical) potentials, semi-empirical approach, non-empirical approach (first principles modeling), and Monte Carlo method [8,9].

The article proposes a new empirical model for determining the thickness of the surface layer and the surface energy of aviation materials. This model allows us to use equations to estimate their surface energy and the thickness of the surface layer. By changing the

chemical composition of materials, it is possible to predict the operational characteristics of advanced aircraft structures. The studies carried out in this work show the role of the surface on the properties of aviation materials.

### 2.2. Determination of the Solid Bodies Surface Energy

In general, experimental determination of the solid bodies surface energy was difficult because their atoms (molecules) were unable to move freely. However, there was an exception; that is, the plastic flow of metals at temperatures close to melting point [9]. The presence of surface tension (energy) could be understood considering the fact that atoms on liquid or solid surface have a greater potential energy than the atoms or ions inside; thus, the surface energy is usually considered as an excess of energy per unit area [10]. The main methods for surface energy experimental determination are [9,10] the zero-creep method, crystals destruction (splitting) method, "neutral drop" method, powder dissolution method, stages of growth and evaporation method, conical sample method, and "healing scratches" method.

Qualitatively, the difference in solid bodies surface energy can be demonstrated with a drop of water spreading ("neutral drop" method) on solid surface (Figure 1).

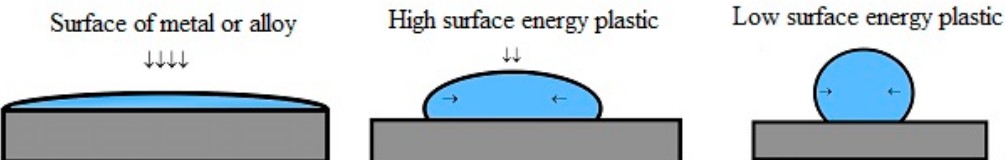

**Figure 1.** Drop of water spreading on solid bodies surfaces.

On metal or its alloy surface, as well as on aircraft materials, the surface energy was obviously greater than that of other materials and the drop spread almost completely.

When a metal surface has significant surface energy, the drop of water spread almost completely, whereas it spread less on high surface energy plastics and hardly spread on low surface energy plastics (Figure 1). Surface energy plays an important role in aviation materials science [11], especially the operational determination of the metal parts surface energy of aviation equipment using a special device for measuring the contact potential difference (Figure 2) [12].

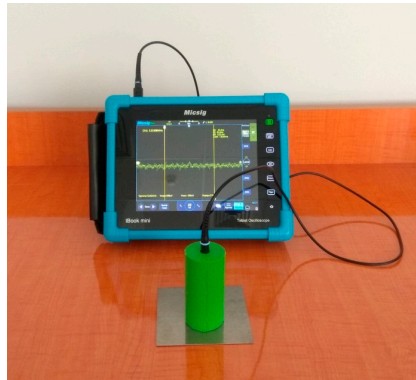

**Figure 2.** Device for measuring surface energy.

In this work, we showed a new atomically smooth crystals surface layer model, in which the roughness of 0.05 nm was neglected. This model was then applied to aircraft materials. To experimentally evaluate the proposed model, we have used a special device that was developed in our laboratories [12].

## 3. Results and Discussion

### 3.1. Features of Aviation Materials Surface Layer

According to modern concepts, the surface layer of metals is a very thin phase that is in thermodynamic equilibrium with the bulk of the metal [2,3]. Various approaches are used to calculate the thickness of the surface layer. For example, the authors of [13] introduce the concept of a natural surface layer of metals, which differs from the base metal based on its physicochemical properties. This surface layer is characterized by high stresses arising within it. Stresses in the surface layer of metal can occur as a result of mechanical, thermal, chemical, and electrochemical processing. The study of such surface layers of metals led to the emergence of a new scientific direction: surface engineering [14]. The main causes of changes in metal surfaces are corrosion, tribology, and destruction of the material. We believe that such a surface layer is correctly known as a technological surface layer. The thickness of such a technological layer ranges from ten to several hundred microns.

### 3.2. Surface Layer Thickness of Pure Metals

It is known [2,3] that the splitting of single crystals in vacuum along the splitting plane can form surfaces consisting of three main types: singular (atomically smooth), vicinal (stepped), and non-singular (diffusion) surfaces (Figure 3).

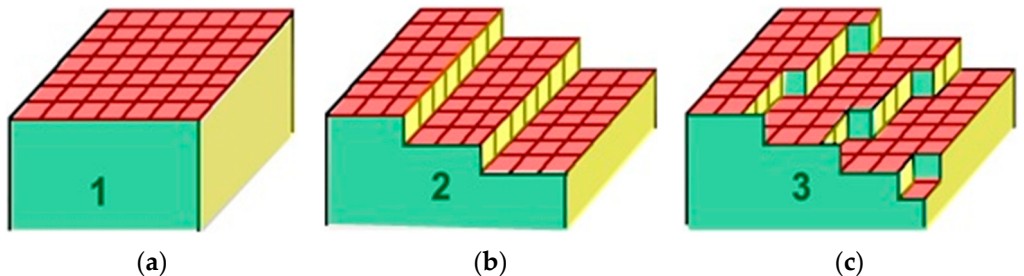

| (a) | (b) | (c) |

**Figure 3.** Three types of surfaces: singular (atomically smooth) (**a**), vicinal (stepped) (**b**), and non-singular (diffusive) surfaces (**c**).

On singular surfaces, the transition from solid phase to vapor phase takes place within one layer, whereas on vicinal surfaces this transition takes place through several crystallographic planes separated via monoatomic steps. Finally, on diffusive surfaces it occurs over several atomic layers. Moreover, the layer thickness is unknown in general. However, the study of such surfaces became possible after the development of ultra-high vacuum technology, atomic force, and tunneling spectroscopy [15]. In our prior work [16], a generalized model of atomically smooth metals surface layer was proposed. This model is shown schematically in Figure 3.

According to our model (Figure 4), the surface layer of an atomically smooth metal consists of two layers: d(I) and d(II). A layer with thickness h from 0 to d will be known as a d(I) layer, while a layer with h ranging from d to 9 d will be known as a d(II) layer (Figure 4a). These two layers periodically change in accordance with the element atomic volume (Figure 4b). In this context, reconstruction and relaxation occur in the d(I) layer with pure metal atoms (Figure 5) as a result of surface rearrangement [2,3].

Moreover, the relaxed surface is only characterized by a change in the interplanar distances, while the reconstructed surface may have a difference in the arrangement of atoms close to the surface (Figure 5). Surface relaxation takes place in most metals. However, reconstructions are observed on the surfaces of some noble and semi-noble face centered cubic metals and transition body centered cubic metals. For gold, the lattice constant is a = 0.41 nm and the surface is rearranged at a distance $d(I)_{Au} = 1.2/0.41 \approx 3$ from three atomic monolayers. Size effects in the d(I) layer are determined by the entire group of atoms in the system [16]. Experimentally, they can be observed on very pure single crystals with grazing incidence X-rays, when the angle of incidence is equal to or less than

the critical angle of total internal reflection [17]. When the angle of incidence becomes less than the critical one, the refracted wave decays exponentially in the volume at a characteristic depth in the order of several nanometers. As a result, a vanishing wave is formed, which propagates parallel to the surface. Therefore, the diffraction of such waves provides information about the surface layer structure [2,3].

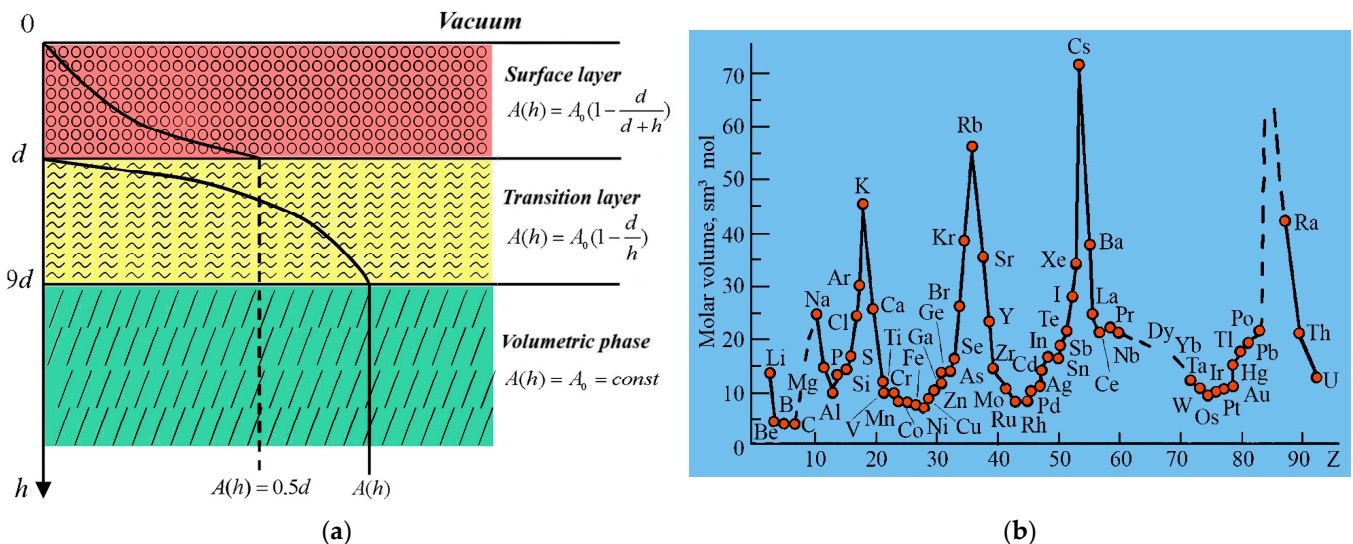

(a)        (b)

**Figure 4.** Schematic representation of surface layer (**a**) and periodic change in elements atomic volume (**b**).

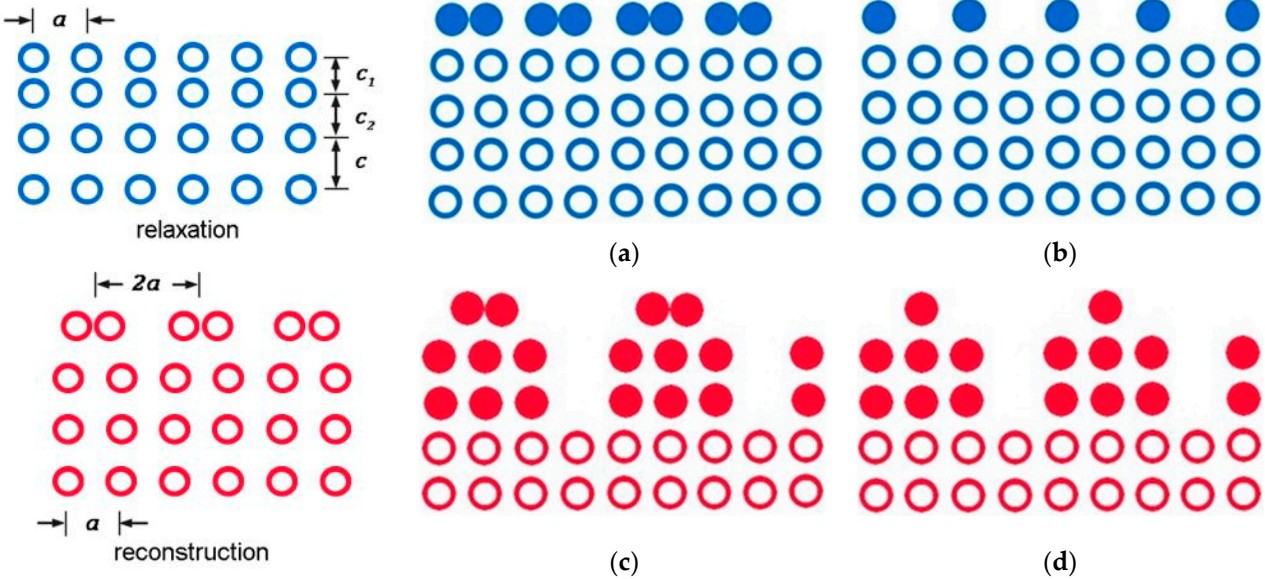

(a)        (b)

(c)        (d)

**Figure 5.** Transformation of metal surface: relaxation—upper layer (**a**,**b**); reconstruction—multiple layers (**c**,**d**).

The d(II) layer extends approximately to $h \approx 9\,d$, where the bulk phase begins. From this dimension (<9 d) the dimensional properties begin. Nanomaterials are commonly understood as materials whose main structural elements do not exceed the nanotechnological boundary of ~100 nm, at least in one direction [17,18]. From the point of view of many researchers, the nanostructure's upper limit (the elements maximum size) should be related to certain critical characteristic parameters: the mean free path of carriers in transport phenomena, the size of domains/domain walls, the diameter of Frank–Read loop for dislocation glide, etc. [17,18]. This means that the d(II) layer should have many size

effects that are related to temperature (Figure 6a), magnetism (Figure 6b), optics (Figure 6c), and other physical properties [18].

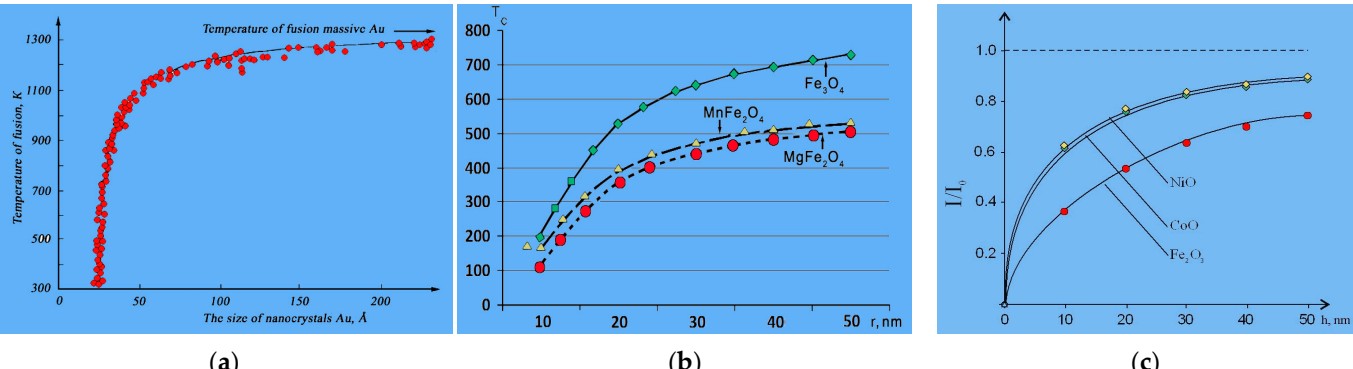

**Figure 6.** Size dependence of Au temperature (**a**), Curie temperature Tc (**b**), and oxide luminescence (**c**).

At h = d, a phase transition takes place in the surface layer (Figure 4a). It is accompanied with sharp changes in the physical properties; for example, the direct Hall–Petch effect is reversed. To describe phase transitions in nanostructures, various models have been proposed; of these models, it is worth mentioning the Landau mean field method, which uses the order parameter. We have used Landau theory [19], albeit replacing the temperature T with the coordinate h. It has been theoretically shown [20] that the heat capacity jump will be $\Delta C_p \equiv 0.5\,d = 1.25$ (Figure 4a) (J/(mol·K)). When conducting the same calculations using the molecular dynamics method [21] of gold heat capacity with particle sizes from 1.5 to 5.5 nm, we have found that $\Delta C_p \approx 1.65$ (J/(mol·K)), which is close to our calculations considering the approximations of computer calculations.

To determine the surface layer thickness h = d(I), we used the melting temperature size dependence (Figure 6a) [22], the Curie temperature for magnets (Figure 6b), and the luminescence intensity of metal oxides (Figure 6c) [23]:

$$A(h) = A_0 \cdot \left(1 - \frac{d(I)}{h}\right),\ h >> d(I)$$
$$A(h) = A_0 \cdot \left(1 - \frac{d(I)}{d(I)+h}\right),\ 0 < h \le d(I). \tag{1}$$

A(h) is the physical property of the surface layer h and $A_0$ is the physical property of a massive sample (crystal), where the properties do not depend on the size (there is no size effect). The physical properties of solid bodies include: melting point, heat capacity, magnetic permeability, and other properties. If in the first Equation (1) we use the melting point T(r) instead of A(h), we will find [24]:

$$T(r) = T_0 \cdot \left(1 - \frac{2\delta}{r}\right),\ \delta >> r, \tag{2}$$

Equation (2) was obtained by Tolman back in 1949 based on Gibbs theory [24]; in Equation (2), r is the solid body sphere radius and δ is the parameter known as Tolman length (by analogy with the de Broglie length), which was not determined via in the Gibbs theory framework and, thus, was considered an unknown parameter.

The parameter d(I) is related to the surface energy σ through the following equation [16]:

$$d(I) = \frac{2\sigma\upsilon}{R_0 T}, \tag{3}$$

σ is the surface energy of the massive sample, υ is the volume of one mole, $R_0$ is the gas constant, and T is the temperature (K).

In our previous works [16,25], we have shown that the following relation is valid:

$$\sigma = \alpha \cdot 0.7 \cdot 10^{-3} \cdot T_m \; [J/m^2] , \tag{4}$$

where the coefficient $\alpha = 1 \; J/m^2 \; K$ and $T_m$ is the solid melting point (K).

Equation (4) is valid for all metals and other crystalline compounds [25]. If we substitute it into (3), at $T = T_m$ we find:

$$d(I) = \beta \cdot 0.17 \cdot 10^{-9} \upsilon \; (m), \tag{5}$$

where the coefficient $\beta = 1 \; (kg/m^3)$ at %·m.

Equation (5) shows that the surface layer thickness of an atomically smooth crystal d(I) is determined via one fundamental parameter—the element atomic volume ($\upsilon = M/\varrho$, M is the molar mass, and $\varrho$ is the element density)—which periodically changes in accordance with the D.I. Mendeleev table (Figure 4b).

Equation (5) allows us to numerically determine the thickness of the surface layer d(I) in several atomic layers, which van der Waals, Guggenheim and Rusanov considered as a layer of finite thickness [1].

The values of the d(I) layer for some cubic metals are given in Table 1, using the following relations [26]:

$$
\begin{aligned}
&\text{Pm3m, } Z = 1, \; l_{100} = 2d(I), \; l_{110} = d(I)\sqrt{2}, \; l_{111} = 2d(I)/\sqrt{3} \\
&\text{Im3m, } Z = 2, \; l_{100} = d(I), \; l_{110} = d(I)\sqrt{2}, \; l_{111} = d(I)/\sqrt{3} \\
&\text{Fm3m, } Z = 4, \; l_{100} = d(I), \; l_{110} = d(I)\sqrt{2}, \; l_{111} = d(I)/\sqrt{3} \\
&\text{Fd3m, } Z = 8, \; l_{100} = d(I)/2, \; l_{110} = d(I)/\sqrt{2}, \; l_{111} = 2d(I)/\sqrt{3},
\end{aligned} \tag{6}
$$

where Z is the number of monolayers.

Table 1 shows the results of our calculations of the thickness of the surface layers d(I) and d(II). The Table 1 [27] shows that the surface layer d(I) thickness is in the range from 1 to 6 nm for all elements except for K, Rb, Cs, Sr and Ba, which means that this layer represents a nanostructure. With d(I), the values in parentheses are calculated via the relation n = d(I)/a (a is the lattice constant) and represent the number of monolayers that exist in the corresponding layer. The layers d(I) and d(II) of cubic crystals have an anisotropy that changes at Z = 2, 4 and 8 in accordance with the Equation (6). The surface layer d(II) thickness for Rb and Cs metals exceeds the Slater technological limit [28], which is 100 nm. Equation (5) shows that the surface layer d(I) thickness can be determined with 5% accuracy not only from experimental data but also, theoretically, using the values of $\upsilon = M/\varrho$ [27,28].

Using Equation (4), as well as the second Equation in (1), we can calculate the surface energy by replacing the term d(I) in Equation (6) with $\sigma_{(hkl)}$.

The d(I) layer experimentally shown in the Table 1 is determined in high vacuum via X-ray scattering. For silicon, d(I) = 3.1 nm, while for gold, d(I) = 2.5 nm [2].

Equation (1) reflects the fact that a phase transition occurs between layers d(I) and d(II) (Figure 5a). It has been theoretically shown [20] that the heat capacity jump for gold is $\Delta Cp = 0.5 \; d = 0.5 \cdot 2.5 = 1.25 \; (J/mol \; K)$. Calculations via the method of molecular dynamics [21] of the heat capacity of gold with particle sizes ranging from 1.5 to 5.5 nm showed that $\Delta Cp \approx 1.65 \; (J/mol \; K)$. This finding is close to our result, given the approximation of computer calculations.

In both our work [20] and in [21], we are talking about a second-order phase transition (Figure 5a), where the heat capacity changes abruptly in the framework of the Landau mean field theory.

**Table 1.** Structure and thickness of surface layer of metals [27].

| Metal | Structure | (hkl) | d(I), nm | d(II), nm |
|---|---|---|---|---|
| Li | Im3m<br>*a* = 0.3502 nm, Z = 2 | (100)<br>(110)<br>(111) | 2.2 (6)<br>3.1 (9)<br>1.3 (4) | 19.8<br>27.9<br>11.7 |
| Na | Im3m<br>*a* = 0.4282 nm, Z = 2 | (100)<br>(110)<br>(111) | 4.5 (11)<br>6.3 (15)<br>2.6 (6) | 40.5<br>56.7<br>23.4 |
| K | Im3m<br>*a* = 0.5247 nm, Z = 2 | (100)<br>(110)<br>(111) | 7.7 (15)<br>10.8 (21)<br>4.5 (9) | 71.1<br>97.2<br>40.5 |
| Rb | Im3m<br>*a* = 0.5710 nm, Z = 2 | (100)<br>(110)<br>(111) | 10.0 (18)<br>14.0 (25)<br>5.9 (10) | 90.0<br>126.0<br>53.1 |
| Cs | Im3m<br>*a* = 0.6141 nm, Z = 2 | (100)<br>(110)<br>(111) | 12.1 (20)<br>16.9 (24)<br>7.1 (12) | 108.9<br>152.1<br>63.9 |
| Ca | Fm3m<br>*a* = 0.5580 nm,<br>Z = 4 | (100)<br>(110)<br>(111) | 4.4 (8)<br>6.2 (11)<br>5.2 (9) | 39.6<br>55.8<br>46.8 |
| Ba | Im3m<br>*a* = 0.5010 nm, Z = 2 | (100)<br>(110)<br>(111) | 6.6 (13)<br>9.2 (18)<br>3.9 (8) | 59.4<br>82.8<br>35.1 |
| Al | Fm3m<br>*a* = 0.4041 nm,<br>Z = 4 | (100)<br>(110)<br>(111) | 1.6 (4)<br>2.2 (6)<br>1.9 (5) | 14.4<br>19.8<br>17.1 |
| Si | Fd3m<br>*a* = 0.5431 nm,<br>Z = 8 | (100)<br>(110)<br>(111) | 1.1 (2)<br>1.5 (3)<br>2.5 (5) | 9.9<br>13.5<br>22.5 |
| Ge | Fd3m<br>*a* = 0.5660 nm,<br>Z = 8 | (100)<br>(110)<br>(111) | 1.2 (2)<br>1.7 (3)<br>2.8 (5) | 10.8<br>15.3<br>25.2 |
| Pb | Fm3m<br>*a* = 0.4950 nm,<br>Z = 4 | (100)<br>(110)<br>(111) | 3.1 (6)<br>4.3 (9)<br>3.7 (7) | 27.9<br>38.7<br>33.3 |
| Cu | Fm3m<br>*a* = 0.3615 nm,<br>Z = 4 | (100)<br>(110)<br>(111) | 1.2 (3)<br>1.7 (5)<br>1.4 (4) | 10.8<br>15.3<br>12.6 |
| Ag | Fm3m<br>*a* = 0.4086 nm,<br>Z = 4 | (100)<br>(110)<br>(111) | 1.7 (4)<br>2.4 (6)<br>2.0 (5) | 15.3<br>21.6<br>18.0 |
| Au | Fm3m<br>*a* = 0.4078 nm,<br>Z = 4 | (100)<br>(110)<br>(111) | 1.7 (4)<br>2.4 (6)<br>2.0 (5) | 15.3<br>21.6<br>18.0 |
| Cr | Im3m<br>*a* = 0.2885 nm, Z = 2 | (100)<br>(110)<br>(111) | 1.2 (4)<br>1.7 (6)<br>0.7 (2) | 10.8<br>15.3<br>6.3 |
| Mo | Im3m<br>*a* = 0.3147 nm, Z = 2 | (100)<br>(110)<br>(111) | 1.6 (5)<br>2.2 (7)<br>0.9 (3) | 14.4<br>19.8<br>8.1 |
| W | Im3m<br>*a* = 0.3160 nm, Z = 2 | (100)<br>(110)<br>(111) | 1.6 (5)<br>2.2 (7)<br>0.9 (3) | 14.4<br>19.8<br>8.1 |

**Table 1.** *Cont.*

| Metal | Structure | (hkl) | d(I), nm | d(II), nm |
|---|---|---|---|---|
| Mn | Im3m $a = 0.8890$ nm, $Z = 2$ | (100) (110) (111) | 1.3 (2) 1.8 (2) 0.8 (1) | 11.7 16.2 7.2 |
| Fe | Im3m $a = 0.2866$ nm, $Z = 2$ | (100) (110) (111) | 1.2 (4) 1.7 (6) 0.7 (2) | 10.8 15.3 6.3 |
| Ni | Fm3m $a = 0.3524$ nm, $Z = 4$ | (100) (110) (111) | 1.1 (3) 1.5 (5) 1.3 (4) | 9.9 13.5 11.7 |
| Ce | Fm3m $a = 0.5160$ nm, $Z = 4$ | (100) (110) (111) | 3.6 (7) 5.0 (10) 4.2 (8) | 32.4 45.0 37.8 |
| Eu | Im3m $a = 0.4581$ nm, $Z = 2$ | (100) (110) (111) | 5.0 (11) 7.0 (15) 2.9 (7) | 45.0 63.0 26.1 |

Table 2 [27] shows that the surface energy σ is anisotropic on different crystal faces and almost two times lower in the d(I) layer compared to the d(II) layer. The results presented in Table 2 were obtained by the authors.

**Table 2.** Surface energy of cubic metals nanostructures.

| Metal | Structure | (hkl) | $\sigma_{(hkl)}$ d(I), J/m$^2$ | $\sigma_{(hkl)}$ d(II), J/m$^2$ |
|---|---|---|---|---|
| Li | Im3m $a = 0.3502$ nm, $Z = 2$ | (100) (110) (111) | 0.159 0.186 0.118 | 0.318 0.445 0.187 |
| Na | Im3m $a = 0.4282$ nm, $Z = 2$ | (100) (110) (111) | 0.137 0.160 0.100 | 0.260 0.364 0.153 |
| K | Im3m $a = 0.5247$ nm, $Z = 2$ | (100) (110) (111) | 0.118 0.138 0.087 | 0.236 0.330 0.139 |
| Rb | Im3m $a = 0.5710$ nm, $Z = 2$ | (100) (110) (111) | 0.109 0.127 0.081 | 0.218 0.305 0.128 |
| Cs | Im3m $a = 0.6141$ nm, $Z = 2$ | (100) (110) (111) | 0.106 0.123 0.078 | 0.211 0.295 0.124 |
| Ca | Fm3m $a = 0.5580$ nm, $Z = 4$ | (100) (110) (111) | 0.389 0.455 0.421 | 0.778 1.089 0.915 |
| Ba | Im3m $a = 0.5010$ nm, $Z = 2$ | (100) (110) (111) | 0.351 0.408 0.260 | 0.701 0.981 0.412 |
| Al | Fm3m $a = 0.4041$ nm, $Z = 4$ | (100) (110) (111) | 0.327 0.379 0.355 | 0.654 0.916 0.769 |
| Si | Fd3m $a = 0.5431$ nm, $Z = 8$ | (100) (110) (111) | 0.495 0.571 0.688 | 0.591 0.844 1.391 |

**Table 2.** *Cont.*

| Metal | Structure | (hkl) | $\sigma_{(hkl)}$ d(I), J/m$^2$ | $\sigma_{(hkl)}$ d(II), J/m$^2$ |
|---|---|---|---|---|
| Ge | Fd3m<br>$a = 0.5660$ nm, Z = 8 | (100)<br>(110)<br>(111) | 0.424<br>0.497<br>0.593 | 0.424<br>0.605<br>0.996 |
| Pb | Fm3m<br>$a = 0.4950$ nm, Z = 4 | (100)<br>(110)<br>(111) | 0.210<br>0.244<br>0.229 | 0.420<br>0.588<br>0.494 |
| Cu | Fm3m<br>$a = 0.3615$ nm, Z = 4 | (100)<br>(110)<br>(111) | 0.475<br>0.557<br>0.512 | 0.950<br>1.330<br>1.118 |
| Ag | Fm3m<br>$a = 0.4086$ nm, Z = 4 | (100)<br>(110)<br>(111) | 0.432<br>0.432<br>0.467 | 0.865<br>1.211<br>1.018 |
| Au | Fm3m<br>$a = 0.4078$ nm, Z = 4 | (100)<br>(110)<br>(111) | 0.468<br>0.548<br>0.430 | 0.936<br>1.310<br>1.101 |
| Cr | Im3m<br>$a = 0.2885$ nm, Z = 2 | (100)<br>(110)<br>(111) | 0.746<br>0.874<br>0.549 | 1.491<br>2.087<br>0.877 |
| Mo | Im3m<br>$a = 0.3147$ nm, Z = 2 | (100)<br>(110)<br>(111) | 1.014<br>1.174<br>0.730 | 2.027<br>2.838<br>1.190 |
| W | Im3m<br>$a = 0.3160$ nm, Z = 2 | (100)<br>(110)<br>(111) | 1.293<br>1.497<br>0.931 | 2.587<br>3.622<br>1.522 |
| Mn | Im3m<br>$a = 0.8890$ nm, Z = 2 | (100)<br>(110)<br>(111) | 0.531<br>0.617<br>0.405 | 1.062<br>1.487<br>0.625 |
| Fe | Im3m<br>$a = 0.2866$ nm, Z = 2 | (100)<br>(110)<br>(111) | 0.634<br>0.743<br>0.467 | 1.268<br>1.775<br>0.746 |
| Ni | Fm3m<br>$a = 0.3524$ nm, Z = 4 | (100)<br>(110)<br>(111) | 0.604<br>0.697<br>0.655 | 1.208<br>1.691<br>1.421 |
| Ce | Fm3m<br>$a = 0.5160$ nm, Z = 4 | (100)<br>(110)<br>(111) | 0.375<br>0.439<br>0.404 | 0.750<br>1.050<br>0.882 |
| Eu | Im3m<br>$a = 0.4581$ nm, Z = 2 | (100)<br>(110)<br>(111) | 0.385<br>0.445<br>0.282 | 0.769<br>1.077<br>0.452 |

In Table 3, we present a comparison between the results obtained from our model (Table 2) and results obtained based on the other existing models [27,28].

Table 3 shows that the results of our model are close to the other calculations within experimental errors. In [30], the surface energy of cubic metals and its anisotropy are calculated using a model related to the co-ordination melting of crystals, whereas in [31] these values are calculated via the Green's function method; in [29], they are calculated via the density functional theory method.

As shown in Table 3, the value of σ(hkl) for tungsten W(111), according to our model, is σ = 1.522 J/m$^{2}$; according to the second model, it is σ = 1.743 J/m$^2$, which represents a slight difference; However, according to the density functional theory method it is equal to σ = 3.939 J/m$^2$. Such a large difference is explained by the peculiarities of this theory. It is

quite difficult to determine the surface energy of a solid body and its anisotropy since the atoms on the surface do not have mobility akin to liquid molecules [32]. Reliable results for the surface energy of metals in the solid state were obtained via the "zero" creep method at a temperature close to the melting point when the creep of atoms proceeds in the diffusion mode. For tungsten W, this method obtained $\sigma$ = 2.690 J/m$^2$ [33] according to our data (Table 3) for tungsten W (100) $\sigma$ = 2.587 J/m$^2$, which differs insignificantly.

**Table 3.** Comparison of massive cubic metals surface energy calculated using different models, including our suggested model.

| Metal | Structure | (hkl) | $\sigma_{(hkl)}$, J/m$^2$ (Counted by the Authors) | $\sigma_{(hkl)}$, J/m$^2$ [29] | $\sigma_{(hkl)}$, J/m$^2$ [30] | $\sigma_{(hkl)}$, J/m$^2$ [31] |
|---|---|---|---|---|---|---|
| Li | Im3m $a$ = 0.3502 nm, Z = 2 | (100) | 0.318 | 0.304 | 0.436 | 0.541 |
| | | (110) | 0.445 | 0.430 | 0.458 | 0.585 |
| | | (111) | 0.187 | 0.180 | - | 0.601 |
| Na | Im3m $a$ = 0.4282 nm, Z = 2 | (100) | 0.260 | 0.189 | 0.236 | 0.258 |
| | | (110) | 0.364 | 0.267 | 0.307 | 0.247 |
| | | (111) | 0.153 | 0.109 | - | 0.302 |
| K | Im3m $a$ = 0.5247 nm, Z = 2 | (100) | 0.236 | 0.124 | 0.129 | 0.148 |
| | | (110) | 0.330 | 0.175 | 0.116 | 0.137 |
| | | (111) | 0.139 | 0.072 | 0.112 | 0.165 |
| Rb | Im3m $a$ = 0.5710 nm, Z = 2 | (100) | 0.218 | 0.101 | 0.107 | 0.126 |
| | | (110) | 0.305 | 0.143 | 0.092 | 0.110 |
| | | (111) | 0.128 | 0.058 | 0.089 | 0.135 |
| Cs | Im3m $a$ = 0.6141 nm, Z = 2 | (100) | 0.211 | 0.085 | 0.092 | 0.114 |
| | | (110) | 0.295 | 0.120 | 0.072 | 0.097 |
| | | (111) | 0.124 | 0.049 | 0.070 | 0.119 |
| Ca | Fm3m $a$ = 0.5580 nm, Z = 4 | (100) | 0.778 | 0.630 | - | 0.529 |
| | | (110) | 1.089 | 0.445 | 0.339 | 0.635 |
| | | (111) | 0.915 | 0.728 | 0.352 | 0.548 |
| Ba | Im3m $a$ = 0.5010 nm, Z = 2 | (100) | 0.701 | 0.365 | - | 0.415 |
| | | (110) | 0.981 | 0.516 | 0.260 | 0.407 |
| | | (111) | 0.412 | 0.211 | 0.258 | 0.495 |
| | | (111) | 0.494 | - | - | - |
| Cr | Im3m $a$ = 0.2885 nm, Z = 2 | (100) | 1.491 | 1.460 | 2.270 | - |
| | | (110) | 2.087 | 2.017 | - | - |
| | | (111) | 0.877 | 2.852 | 3.090 | - |
| Mo | Im3m $a$ = 0.3147 nm, Z = 2 | (100) | 2.027 | 2.306 | - | 3.661 |
| | | (110) | 2.838 | 3.261 | 3.180 | 3.174 |
| | | (111) | 1.190 | 1.331 | 2.500 | 3.447 |
| W | Im3m $a$ = 0.3160 nm, Z = 2 | (100) | 2.587 | 3.020 | - | 4.403 |
| | | (110) | 3.622 | 4.270 | 3.840 | 3.649 |
| | | (111) | 1.522 | 1.743 | 2.500 | 3.939 |

For the destruction specific energy of material (metal), it is necessary to consider the absorbed energy level at which this material reaches complete exhaustion of its resistance to external forces, i.e., the complete loss of its structure-bearing capacity. According to E. Orovan [34], the intensity of material destruction is determined via the adhesion energy at the phase boundary:

$$W(hkl) = 2\sigma_{(hkl)} \ [J/m^2] . \tag{7}$$

Since the surface energy of the d(I) layer is two times lower than the surface energy of the d(II) layer and the bulk phase, the destruction of metal starts from the surface. Moreover, for a metal with Z = 2 the destruction begins from the (111) face, while for Z = 4 it begins

8 from the (100) face. The corrosion of aircraft parts (especially steel and aluminum) begins from the surface due to various weather conditions. Therefore, the strengthening of aircraft parts and improvement of their anti-corrosion properties should begin with the d(I) layer.

Thus, the model proposed is universal since the thickness of the surface layer d(I) presented in the Table 1 is comparable to the size of the nanocrack L, which arises due to stresses caused by relaxation or reconstruction of the surface (Equation (7) and Table 3) and determines the destruction of aviation materials and, in general, any structural materials.

If we take into account the size of nanocracks (Table 1), their formation should occur in a few ns. This approach was recently demonstrated in [35,36].

The method presented in [35,36] is based on fractoluminescence as, during the destruction of a solid, a light signal (luminescence) occurs when atomic bonds are broken on the surface of nanocracks with a time resolution of 1 to 2 ns. In [36], the fractoluminescence spectrum of oligoclase was obtained upon destruction of its surface. The duration of the signals was about 50 ns, while the time interval between them varied from 0.1 to 1.0 μs. The spectrum contained four maxima, which appeared when dislocations overcame four barriers along slip planes. In this case, dislocations form primary cracks with a size of about 10 to 20 nm. Oligoclase [35] is a mixture of 10–30% anorthite $CaAl_2Si_2O_8$ and 70–90% albite $NaAlSi_3O_8$. The calculation via Equation (5) gave L = d(I) = 16.8–17.2 nm, which is in good agreement with the experiment.

### 3.3. Surface Layer Thickness and Surface Energy of Aviation Materials

Nickel aluminides are identified by their high melting points, low density (Table 4 [37]), heat resistance when oxidized in air up to 1200 °C, and high resistance to thermal shocks. The high resistance of nickel-based aluminides to oxidation is the reason for their wide use as protective coatings for the parts of high-tech products in aerospace and power engineering, including the gas turbine plants elements and rocket engines for various purposes [37].

**Table 4.** Thickness of surface layer and surface energy of aluminides [34].

| Aluminide | Molar Mass, mol$^{-1}$ | Density, g/cm$^3$ | Melting Point, K | d(I), nm | σ, J/m$^2$ |
|---|---|---|---|---|---|
| $NiAl_3$ | 139.65 | 3.957 | 1127 | 8.5 (13) a = 0.661 | 0.879 |
| $Ni_2Al_3$ | 174.42 | 4.787 | 1406 | 8.7 (22) a = 0.4036 | 1.097 |
| $Ni_3Al$ | 202.84 | 7.293 | 1668 | 6.7 (19) a = 0.3589 | 1.301 |

The $NiAl_3$ phase has an orthorhombic lattice, whereas the $Ni_2Al_3$ phase, on the basis of which the solid solution is formed, has a hexagonal lattice. In the Ni-Al system, the γ′-phase is in equilibrium with the nickel-based γ-solid solution, which is a solid solution based on the $Ni_3Al$ intermetallic compound. The intermetallic compound $Ni_3Al$ belongs to the alloys with the L12 superlattice, which are characterized by high ordering energy. Ordered atoms form crystal lattices, which can be represented as several sub-lattices built into one another. The fundamental difference between the γ′-phase and a solid solution of aluminum in nickel is the presence of an ordered structure up to the melting point [38].

At present, intensive development of new ductile alloys that have high ability for deformation in the cold state is based on β-modification of titanium. These alloys allow the production of high-strength sheets and plates, which are used in manufacturing airframe skin parts and power sets for new aircraft [39] based on a high-entropy alloy, such as CuNiAlFeCr [40–43]. By analyzing Tables 4 and 5 [39], we can be see that by increasing molar mass and decreasing density, higher performance properties can be obtained compared to similar compounds.

The presented model is empirical but very important since, even for all atomically smooth metals, the size of the surface layer is determined in high vacuum [2].

**Table 5.** Surface layer thickness and surface energy of β-modification of titanium and high-entropy alloy CuNiAlFeCr [39].

| Alloy | Molar Mass, mol$^{-1}$ | Density, g/cm$^3$ | Melting Point, K | d(I), nm | σ, J/m$^2$ |
|---|---|---|---|---|---|
| Ti + 0.03Al + 0.15V + 0.03Sn + 0.03Cr + 0.015Zr + 0.015Mo | 64.28 | 4.774 | 1033 | 3.2 | 0.806 |
| Ti + 0.03Al + 0.06V + 0.11Cr + 0.015Mo + Zr | 157.46 | 4.890 | 993 | 7.7 | 0.775 |
| CuNiAlFeCr | 257.10 | 7.165 | 1369 | 8.6 | 1.068 |

## 4. Conclusions

In this work, a new empirical model for theoretically calculating the surface layer thickness and surface energy of aviation metallic materials is proposed. The model presents equations that allow the assessment of the most important nanostructure characteristics of aviation materials: surface energy σ (J/m$^2$) and the thickness of surface layers d(I) and d(II) (nm). In the normal state in air, metals are always covered, depending on the processing modes, with an oxide layer with a thickness of 5 to 90 nm. By changing the chemical and elemental composition of aviation materials using simple software calculations, it is possible to predict the performance characteristics of advanced aircraft structures using the equations presented in this work. To prove the feasibility of this model, a comparison with the existing models is presented. Moreover, based on the proposed model, the surface layers of pure metals are calculated theoretically for the first time; it has been shown that this layer may be regarded as a nanostructure with a size of 3–5 nm. When applying the same model to the surface layer of aviation metallic materials, the calculations showed that it is a nanostructure with a size of 6–9 nm. In these regards, the surface layer of aircraft should contain a nanostructure with a size of 70–90 nm. The studies carried out in this work show the role of the surface thickness on the physical properties of aviation metallic materials.

**Author Contributions:** Conceptualization, V.M.Y. and V.I.G.; methodology, V.S.O. and A.V.R.; software, V.I.G.; validation, V.M.Y., V.I.G., V.S.O. and A.V.R.; formal analysis, A.V.R.; investigation, V.S.O.; resources, A.V.R.; data curation, V.M.Y.; writing—original draft preparation, V.M.Y., V.I.G. and V.S.O.; writing—review and editing, V.M.Y., V.I.G. and V.S.O.; visualization, V.M.Y.; supervision A.V.R.; project administration, A.V.R.; funding acquisition, V.S.O. All authors have read and agreed to the published version of the manuscript.

**Funding:** The work was performed in accordance with program of the Ministry of Education and Science of the Republic of Kazakhstan. Grants No. 0118RK000063 and No. F.0781. of the Ministry of Education and Science of the Republic of Kazakhstan (MES RK).

**Data Availability Statement:** Not applicable.

**Conflicts of Interest:** The authors declare no conflict of interest.

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
