# Peer review of "Calculating the Surface Layer Thickness and Surface Energy of Aircraft Materials"

_inventions, doi:10.3390/inventions8030066_

Round 1

Reviewer 1 Report (New Reviewer)

1. The author includes recent references for better understanding for beginners.

2. More discussion is required in all sections, and also include proper citations in the respective results.

3. Initially, the English of the whole chapter requires substantial/careful editing. Grammar and text coherence need a substantial review.

4. The author includes optimum process parameters to achieve better results in the abstract as well as the conclusion part.

5. The results and discussion part could be improved.

Author Response

Comment: More discussion is required in all sections, and also include proper citations in the respective results.
Answer: Our work has an original style. We have added relevant links in the text.

Comment: The results and discussion part could be improved.
Answer: The results and the discussion part were changed.

Comment: Initially, the English of the whole chapter requires substantial/careful editing. Grammar and text coherence need a substantial review.

Answer: The language of article has been revised.

Reviewer 2 Report (New Reviewer)

The Authors propose a new empirical method for the determination of the surface layer thickness and surface energy of materials, comparing its results with those of other existing methods.

The paper content is interesting and also its form is very good.
Therefore, I suggest its publication, with some clarifications which I suggest in the following.

1) line 46: substitute "presents" with "represents" (if this is correct)
2) lines 164-165: does the layer d(I) extend from 0 to d and the layer d(II) extend from d to 9d? You should reword this sentence in order to clarify this point
3) lines 177-178: even though very basic acronyms, you should explicitly write that "FCC" is for "Face Centered Cubic" and "BCC" is for "Body Centered Cubic"
4) have the data in Figure 6 been obtained here or have they been reported from a previous article? In case, this previous article should be cited
5) line 206: in the equation "Delta C_p=0.5d=1.25 J/(mol K)" you should clarify what value is "d" (does it represent how many nanometers are in d?). Moreover, I think you should enclose "mol K" among brackets
6) in the second of equations (1) please correct the unequality signs
7) line 225: add something like "we have shown" before "that"
8) first of equations (6): add a comma before l_{110}
9) line 253: you say that you should replace d(I) with sigma_{hkl}; shouldn't you use equation (3) instead?
10) line 258: I suggest to add a central dot between 0.5 and 2.5
11) lines 273-276: in Table 3 you cite references [29], [30,] [31] but then in the sentences which follow this table you cite refences [32] and [33]; it is true that Refs. [29] and [32], and [31] and [33] have Authors in common (I suppose Bokarev and Bochkarev is the same Author), but I think that in these sentences you should add also the same references cited in the table
12) line 349: modify "most nanostructures important characteristics" to "most important nanostructure characteristics"

Author Response

Comment: line 46: substitute "presents" with "represents" (if this is correct).

Answer: “Presents” has been changed to “represents”.

Comment: lines 164-165: does the layer d(I) extend from 0 to d and the layer d(II) extend from d to 9d? You should reword this sentence in order to clarify this point.
Answer: We have paraphrased the sentence on lines 164–165.

Comment: lines 177-178: even though very basic acronyms, you should explicitly write that "FCC" is for "Face Centered Cubic" and "BCC" is for "Body Centered Cubic".

Answer: The Abbreviations have been deciphered.

Comment: have the data in Figure 6 been obtained here or have they been reported from a previous article? In case, this previous article should be cited.
Answer: The Figure 6 was made by authors on the basis of [18]. Reference to work [18] is inserted into the text.

Comment: line 206: in the equation "Delta C_p=0.5d=1.25 J/(mol K)" you should clarify what value is "d" (does it represent how many nanometers are in d?). Moreover, I think you should enclose "mol K" among brackets.
Answer: The equation ΔСp ≡ 0.5d = 1.25 (Figure 4a) (J/(mol K)) was refined by putting “mol K” in brackets.

Comment: in the second of equations (1) please correct the unequality signs.
Answer: We have corrected the inequality sign.

Comment: line 225: add something like "we have shown" before "that".

Answer: “We have shown” has been added.

Comment: first of equations (6): add a comma before l_{110}.

Answer: Comma has been added.

Comment: line 253: you say that you should replace d(I) with sigma_{hkl}; shouldn't you use equation (3) instead?
Answer: Equation (3) is a general equation relating d(I) and σ. More accurate results are obtained by our method using equations (1) and (4).

Comment: line 258: I suggest to add a central dot between 0.5 and 2.5.

Answer: Central dot between 0.5 and 2.5 has been added.

Comment: lines 273-276: in Table 3 you cite references [29], [30,] [31] but then in the sentences which follow this table you cite refences [32] and [33]; it is true that Refs. [29] and [32], and [31] and [33] have Authors in common (I suppose Bokarev and Bochkarev is the same Author), but I think that in these sentences you should add also the same references cited in the table.

Answer: References citing has been corrected.

Comment: line 349: modify "most nanostructures important characteristics" to "most important nanostructure characteristics"

Answer: The sentence has been modified.

Reviewer 3 Report (New Reviewer)

In my opinion, the article is not printable in its current form. The authors are trying to transfer the results obtained for pure crystals to real construction materials in an unauthorized manner. Important notes:

1.     The main problem in this work is the uncritical transfer the results
of pure crystal calculations to real construction materials. Comparing
the parameters of the surface layer of a pure, single crystal, with the
parameters of the technological surface layer is not justified.
2.     It is difficult to call of the analysed  layer of the material as a surface
  layer with a thickness of (d1 + dII). There is no justification for how
many atomic layers influenced on  the total value of surface free energy.
Calculated values ​​of surface free energy (sfe) on the basis of relation 4
raise doubts. They are much larger than those determined in experimental
studies of technological surface layer. In these tests, the value of surface
free energy usually does not exceed 0.1 J/m2. Presented in Tab. 5 sfe
values ​​exceed at least 10 times the results obtained in experiments for
the technological surface layer.
I take into account the fact of the presence
of a physisorption layer on the macroscopic surface.
3.     I do not question the results those presented in this paper sfe (tab. 1, 2)
values. Theoretical physicists should verify the correctness of these
calculations. However, it seems that basing calculations solely on the
molar volume of the material and melting point is very risky.
4.     The sfe value of metals and their alloys  depends, in real conditions, on
so many factors, including the technological history of the surface layer,
that an attempt to transfer the presented results to industrial practice is
meaningless.
5.  If the calculated theoretical values ​​refer to the specific free surface energy,
then for a geometrically developed surface the actual wetting surface is
larger than the nominal (unit) surface.
This is confirmed by frequent
cohesive failures of the adhesive layer in destructive tests of adhesive
joints. The process is complex because in the conditions of measurement
of the technological surface layer, we are dealing with a physisorption
layer that reduces the sfe value.
  In my opinion, the work should be redrafted. All suggestions as to the possibility
of their practical use should be removed from it.

Author Response

Comment: The main problem in this work is the uncritical transfer the results of pure crystal calculations to real construction materials. Comparing the parameters of the surface layer of a pure, single crystal, with the parameters of the technological surface layer is not justified.
Answer: The universal model developed by authors, based on the properties of ideal crystals, gives results that are in good agreement with the results obtained by other researchers, and can be applied to real surfaces. This is especially justified in relation to nanomaterials, which are increasingly being used.

Comment: It is difficult to call of the analysed  layer of the material as a surface layer with a thickness of (d1 + dII). There is no justification for how many atomic layers influenced on  the total value of surface free energy. Calculated values of surface free energy (sfe) on the basis of relation 4 raise doubts. They are much larger than those determined in experimental studies of technological surface layer. In these tests, the value of surface free energy usually does not exceed 0.1 J/m2. Presented in Tab. 5 sfe values exceed at least 10 times the results obtained in experiments for the technological surface layer. I take into account the fact of the presence of a physisorption layer on the macroscopic surface.
Answer: For the metals presented in the work, the results obtained by authors within the framework of the model used gave such results. Our results are in good agreement with the results obtained by other authors in [29–31, 35, 40].

Comment: I do not question the results those presented in this paper sfe (tab. 1, 2) values. Theoretical physicists should verify the correctness of these calculations. However, it seems that basing calculations solely on the molar volume of the material and melting point is very risky.
Answer: The results of calculations of the surface energy of metals, obtained on the basis of the universal model proposed by us, give good results, consistent with the previously obtained results.

Comment: If the calculated theoretical values refer to the specific free surface energy, then for a geometrically developed surface the actual wetting surface is larger than the nominal (unit) surface. This is confirmed by frequent cohesive failures of the adhesive layer in destructive tests of adhesive joints. The process is complex because in the conditions of measurement of the technological surface layer, we are dealing with a physisorption layer that reduces the sfe value. In my opinion, the work should be redrafted. All suggestions as to the possibility of their practical use should be removed from it.
Answer: Our proposals on the possibility of practical use of the results obtained in the work on lines 363–366 have been deleted by authors.

Round 2

Reviewer 1 Report (New Reviewer)

Accept in present form

Reviewer 3 Report (New Reviewer)

The authors did not answer some of my doubts. The article has been
slightly improved. The authors' conclusions about the possibility
of using the proposed dependencies to forecast the energy state of the
technological surface layer do not convince me. However, I do not oppose
the publication of this work. It can provoke an interesting discussion
and that is also important.
Yours faithfully

This manuscript is a resubmission of an earlier submission. The following is a list of the peer review reports and author responses from that submission.

Round 1

Reviewer 1 Report

acceptable now

Author Response

Dear sirs, thank you very much!

Reviewer 2 Report

In the manuscript, the authors propose an empirical model to predict the 'surface layer thickness' and surface energy of metals, with special application to avionic materials. 

According to the authors, an empirical model is proposed , which "can be easily used for computer simulation of for any cases that rise in materials science without experimental verification" (line 37). 

In my opinion the paper has some serious methodologic flaws, which do not qualify it for publication: 

1. As I understand it, the authors claim to have a universal model of materials science which can replace experiments. However, each model in natural sciences should be tied to experimental verification. There is no model that can replace the experiment, especially not in materials science. Since material fatigue with its serious consequences in aviation is a much more complicated process deserving permenant control, it should not be the aim to replace experimental investigation. I have my doubts that the properties of an avionic material should be entirely determined by its surface properties.  

2. The collection of thoughts presented in the manuscript seems to be inconsistent in itself. Given that eq. (1) of the manuscript is meaningful within the model, the "jump" of heat capacity at the surface of 1.25 J/(mol K) claimed in line 202 is inconsistent.  

3. The said empirical model deals with an atomically smooth surface (line 157-159). "Real" surfaces in nature and in technology, however, have rough surfaces with defects. This "material gap" -- as it is called in surface science -- should allways be kept in mind as a systematic source of error, if one compares model predictions with experiments. 

4. The authors interchange freely results from droplets/nanoparticles with those of a twodimensional flat surface extending in half-space. Why should it be justified to replace the surface property A(h) with the radius dependent melting point T(r) of a droplet (line 213)? 

5. Given that eq. (3) and eq. (4) would be meaningful within the author's model, why should it be justified to equate the temperature T in (3) with the melting temperature T_m in eq. (4) to obtain the relation eq. (5)? In that case all quantities evaluated with the model (table 2, table 3) should refer to the melting point of the considered material. 

6. Table 2: The authors state that the values obtained are 'close to the other calculations within experimental errors' (line 255). However, there are quite large discrepancies, e. g.  for W(111) with sigma values between 1.522 and 3.939 J/m^2. This should be critically discussed. 

Author Response

Comment 1: As I understand it, the authors claim to have a universal model of materials science which can replace experiments. However, each model in natural sciences should be tied to experimental verification. There is no model that can replace the experiment, especially not in materials science. Since material fatigue with its serious consequences in aviation is a much more complicated process deserving permenant control, it should not be the aim to replace experimental investigation. I have my doubts that the properties of an avionic material should be entirely determined by its surface properties.  

Answer 1: The model proposed by us is indeed universal, since no one has yet proposed a method for determining the thickness of the surface layer by a simple method of any connections. We compare this thickness d(I) with a nanocrack L, which arises due to stresses caused by surface relaxation or reconstruction (Eq. (7) and Table 3) and determines the destruction of aviation materials and, in general, any structural materials.

If we take into account the size of nanocracks (Table 1), then their formation should occur within a few nanoseconds. It is this approach that was recently demonstrated in [1, 2].

This method is based on fractoluminescence, when a light signal (luminescence) appears during the destruction of a solid body when atomic bonds are broken on the surface of nanocracks with a time resolution of 1 to 2 nanoseconds (ns). In [2], the fractoluminescence spectrum of oligoclase was obtained upon destruction of its surface. The duration of the signals was ≈ 50 ns, and the time interval between them varied from ≈ 0.1 to 1 µs. The spectrum contained 4 maxima, which appeared when dislocations overcame 4 barriers along slip planes. In this case, the dislocations form primary cracks about 10 to 20 nm in size. Oligoclase is a mixture of 10-30% anorthite CaAl2Si2O8 and 70-90 % albite NaAlSi3O8. The calculation by formula (5) gave L=d(I)=16.8–17.2 nm, which is in good agreement with the experiment:

  • Vettegren, V.I.; Ponomarev, A.V.; Kulik. V.B., Mamalimov, R.I., Shcherbakov, I.P. Destruction of quartz diorite at friction. Geo-Phys. Res. 2020, 21(4), 35–50.
  • Vettegren, V.I.; Ponomarev, A.V.; Mamalimov, R.I.; Shcherbakov, I.P. Nanocracks upon Fracture of Oligoclase. , Phys. of the Sol. Earth 2021, 57(6), 894–899.

Comment 2: The collection of thoughts presented in the manuscript seems to be inconsistent in itself. Given that eq. (1) of the manuscript is meaningful within the model, the "jump" of heat capacity at the surface of 1.25 J/(mol K) claimed in line 202 is inconsistent.  

Answer 2: Van der Waals, Guggenheim, and Rusanov considered the surface layer as a layer of finite thickness several atomic layers in size, but we give it numerically in equation (5). Experimentally, the d(I) layer is determined in high vacuum by X-ray scattering: for silicon d(I)=3.1 nm, for gold d(I)=2.5 nm (see Oura, K.; Lifshits, V.G.; Saranin, А.А.; Zotov, А.V.; Katayama, М. Introduction to surface physics; Science: Moscow, Russia, 2006).

Equation (1) reflects the fact that a phase transition occurs between layers d(I) and d(II) (Figure 4a). It has been theoretically shown [20] that the heat capacity jump for gold is ΔСp = 0.5d = 0.5x2.5 = 1.25 (J/mol K). Calculations by the method of molecular dynamics [21] of the heat capacity of gold with particle sizes from 1.5 to 5.5 nm showed that ΔСp ≈ 1.65 (J/mol·K). This is close to our result, given the approximation of computer calculations.

Both in our work [20] and in work [21], we are talking about a second-order phase transition (Figure 4a), where the heat capacity changes abruptly in the framework of the Landau mean field theory.

Comment 3: The said empirical model deals with an atomically smooth surface (line 157-159). "Real" surfaces in nature and in technology, however, have rough surfaces with defects. This "material gap" -- as it is called in surface science -- should allways be kept in mind as a systematic source of error, if one compares model predictions with experiments. 

Answer 3: For most metals, the roughness and atomically smooth surface is determined by the Jackson criterion α = Lm/R T0, where Lm is the molar heat of crystallization; T0 is the equilibrium crystallization temperature; R is the gas constant. (Jackson K.A. Interface kinetics, in Growth and Perfection of Crystals. // John Wiley, New York. - 1958. - P. 319-324.), if α < 2, then the surface is considered rough, and if α > 2, then the surface is assumed to be atomically smooth. From the Table 1, 4 and from the handbook it follows that in our case α > 2 and the surface is assumed to be atomically smooth.

Comment 4: The authors interchange freely results from droplets/nanoparticles with those of a twodimensional flat surface extending in half-space. Why should it be justified to replace the surface property A(h) with the radius dependent melting point T(r) of a droplet (line 213)? 

Answer 4: A drop of radius r in the Figure 1 of the article turns into a surface with thickness h ≈ r due to Young's equality:

σ31 = σ23 + σ12cos θ

In this case, σ31 – σ23 > 0 and σ31 – σ23 > σ12. Then this condition cannot be satisfied, since the cosine cannot be greater than one. In this case, the liquid spreads over the surface (kerosene or gasoline on the surface of tin, glass), that is, the liquid completely wets the surface of the solid.

Comment 5: Given that eq. (3) and eq. (4) would be meaningful within the author's model, why should it be justified to equate the temperature T in (3) with the melting temperature T_m in eq. (4) to obtain the relation eq. (5)? In that case all quantities evaluated with the model (table 2, table 3) should refer to the melting point of the considered material. 

Answer 5: The melting temperature Tm from (4) as a parameter decreases at Т = Tm in (3) and the thickness d(I) depends according to equation (5) on the molar volume υ = M/ρ, which depend on temperature.

Comment 6: Table 2: The authors state that the values obtained are 'close to the other calculations within experimental errors' (line 255). However, there are quite large discrepancies, e. g.  for W(111) with sigma values between 1.522 and 3.939 J/m^2. This should be critically discussed.

Answer 6: The value for tungsten W (111) according to our model is σ = 1.522 according to the second σ = 1.743 J/m2, which differs insignificantly, but by the density functional theory method it is equal to σ = 3.939 J/m2. These are all the costs of theory. Determining the surface energy of a solid and its anisotropy is rather difficult, since the atoms on the surface do not have the mobility of liquid molecules. Reliable results of the surface energy of metals in the solid state were obtained by "zero" creep method at a temperature close to the melting point, when the creep of atoms proceeds in the diffusion mode. For tungsten W, this method obtained σ = 2.690 J/m2 (Khokonov, Kh.B.; Taova, T.M.; Alchagirov, B.B. Surface energy and surface tension of metals and their binary metal alloys in a solid state. Proc. of the Kabardino-Balkarian State Un. 2019, 9(2), 5-19), according to our data (Table 3) for tungsten W (100) σ = 2.587 J/m2, which also differs slightly.

Round 2

Reviewer 2 Report

In the updated version of the manuscript, the authors have made only slight modifications of the text concerning the validity of their eq. (5) which connects a quantity d(I) to the molar volume of a material. Still it is not clear for the reader why T=Tm (line 228) should be justified. Moreover, eqs. (3), (4), and (5), although they can be found in ref. 16 at least in a similar form, are not derived there, as far as I can see. So, as long as the authors can not provide a clear presentation of their model which the reader can assess, I would doubt its validity. 

The presented model relates surface tension to depend only on the molar volume of a material, not on its chemical properties, e. g. encoded in force-fields or electronic structure theory. Still the authors haven't provided a critical discussion of their model. It is in my opinion audacious and not sufficient to explain the deviating results of different models with "pecularities of the theory" (line 282), in this case density functional theory (which is not ref. 29, as stated in line 277, but apparently ref. 31).